# NanoMoE: Scaling Mixture of Experts to Individual Layers for Parameter-Efficient Deep Learning

## Abstract

Large language models (LLMs) have achieved remarkable success, but their growing size leads to significant challenges in efficiency and cost. This work explores parameter-efficient deep learning, aiming to achieve comparable performance with fewer parameters and floating-point operations (FLOPs). We introduce NanoMoE, a novel family of parameter-efficient building blocks inspired by the Mixture of Experts (MoE) framework. NanoMoE offers a modular and efficient replacement for fully connected layers within traditional neural networks. We instantiate NanoMoE with three variants of increasing complexity and theoretically demonstrate its superior expressivity compared to low-rank factorization with minimal parameter increase. Empirical results validate that NanoMoE achieves superior model quality compared to low-rank factorization under the same parameter or FLOP budget, confirming its enhanced efficiency.

## 1 Introduction

Large language models (LLMs) have demonstrated exceptional performance (Brown et al., 2020; Devlin et al., 2018), yet they still exhibit limitations in factual accuracy (Ji et al., 2023), logical reasoning (Teng et al., 2023), and mathematical proficiency (Collins et al., 2024). The pursuit of ever-increasing model size to overcome these limitations, as seen in the progression from GPT-3 (175B parameters) (Brown et al., 2020) to PaLM (540B parameters) (Chowdhery et al., 2023) and GPT-4 (estimated at 1.8 trillion parameters) (Achiam et al., 2023), leads to significant challenges in parameter efficiency, training efficiency, and inference costs. These challenges are further amplified in multimodal models (Baltrušaitis et al., 2018), where diverse application scenarios demand complex and computationally expensive architectures.

This trend, however, raises a crucial question: **can we achieve comparable performance and learning capacity with a significant reduction in parameters and floating-point operations (FLOPs)**? The pursuit of parameter and FLOP efficiency is paramount due to several critical factors. Firstly, reducing the number of parameters directly translates to lower memory requirements, enabling the deployment of LLMs on resource-constrained devices and reducing the financial burden of model storage (Xu et al., 2024). Secondly, minimizing FLOPs lowers computational costs, leading to faster inference times, decreased energy consumption, and a reduced carbon footprint (Strubell et al., 2020). This efficiency is essential for real-time applications, accessibility, and environmental sustainability. Finally, by optimizing model size and computational complexity, we can promote wider accessibility, enabling researchers and developers with limited resources to leverage the power of LLMs, fostering innovation and broader participation in the field.

Addressing the escalating computational demands of LLMs necessitates the design of parameter-efficient building blocks. While parameter-efficient fine-tuning (PEFT) (Ding et al., 2023; Han et al., 2024) has garnered considerable attention for adapting pre-trained models by optimizing injected adapters (Rebuffi et al., 2018; Houlsby et al., 2019), such as the low-rank adaptation (LoRA) method that injects low-rank factorized adapters into dense layers (Hu et al., 2022), the need for increased efficiency extends to the entire training process, including pre-training. Rather than solely focusing on parameter-efficient adapters injected alongside existing layers, we propose exploring parameter

efficiency within the original layers of the pre-trained model, enabling enhanced learning capacity during the pre-training stage. We formally define this problem as **parameter-efficient deep learning**.

To address this problem, we introduce NanoMoE, a novel neural network structure inspired by the Mixture of Experts (MoE) framework. MoE draws inspiration from real-world problem-solving, where complex issues often necessitate specialized expertise. MoE models utilize "experts," specialized sub-models focusing on specific knowledge areas, with a gating network intelligently routing input queries to the most relevant experts. This facilitates efficient model capacity utilization and adaptability across diverse tasks. While the MoE concept originated in the early 1990s (Jacobs et al., 1991), recent advancements, such as the sparse MoE layer introduced by Shazeer et al. (2016), have revitalized its application in large-scale models. As LLMs continue to grow and application scenarios become more specialized, MoE offers a compelling pathway to address both general and domain-specific tasks within a unified framework, proving particularly valuable for multimodal models handling diverse data and feature relationships. The success of models like Mistral 8x7B (Jiang et al., 2024), which outperforms the larger Llama 2 (Touvron et al., 2023) with fewer parameters, underscores the potential of MoE in achieving comparable or superior performance with reduced computational resources. While recent LLMs like Mistral 8x7B employ MoE to combine large sub-models, NanoMoE is designed as a modular and efficient replacement for fully connected layers within traditional neural networks. This granular approach allows for the integration of multiple NanoMoE blocks within a single model, potentially yielding significant gains in performance and flexibility without a dramatic increase in parameter count.

Our work makes the following contributions:

- We propose NanoMoE, a novel family of parameter-efficient building blocks for neural networks inspired by the MoE framework. We instantiate NanoMoE with three variants: NanoMoE-I, NanoMoE-II, and NanoMoE-III, offering increasing levels of complexity and computational cost.

- We theoretically demonstrate that NanoMoE offers strictly greater expressivity compared to low-rank factorization while requiring only a minimal increase in parameters.

- We empirically validate that NanoMoE achieves superior model quality compared to low-rank factorization. Given a budget of parameters or FLOPs, we compare the train and test loss of NanoMoE against low-rank factorization and observe that NanoMoE consistently demonstrates superior performance, confirming its enhanced parameter and FLOP efficiency.

The remainder of this paper is organized as follows. Section 2 reviews related work. Section 3 presents our proposed NanoMoE method and its theoretical guarantees. Section 4 presents our experimental results. Finally, Section 5 concludes the paper, and Section 6 discusses limitations and future work.

## 2 RELATED WORK

Eigen et al. (2013) proposed stacking MoE layers in a neural network, with the aim of achieving an exponential number of experts as a function of the network depth. Lepikhin et al. (2021) replace every other feed-forward network layer in the Transformer encoder and decoder with a position-wise MoE layer. The Switch Transformer (Fedus et al., 2022) integrates the MoE design into the T5 model and pre-trains it on the C4 dataset, resulting in a fast and effective pre-trained large model. The key innovation of the Switch Transformer is its simplified MoE routing algorithm, which significantly enhances computational efficiency. GLaM (Du et al., 2022) is three times larger than GPT-3; however, due to its use of a sparse MoE design, the training cost is only one-third that of GPT-3, and it outperforms GPT-3 on 29 NLP tasks.

Rebuffi et al. (2018) and Houlsby et al. (2019) propose transferring a model to new tasks by inserting small, task-specific modules, termed *adapter layers*, within the pretrained model's layers. Hu et al. (2022) propose Low-Rank Adaptation (LoRA), which freezes the pre-trained model weights and integrates trainable low-rank factorization matrices into each layer of the large language model. Edalati et al. (2022) and He et al. (2023) utilize the Kronecker product to reparameterize adapter layers for parameter-efficient fine-tuning. Similarly, Mahabadi et al. (2021) introduce the Compacter layer, which builds upon LoRA by inserting a GeLU non-linearity (Hendrycks & Gimpel, 2016)

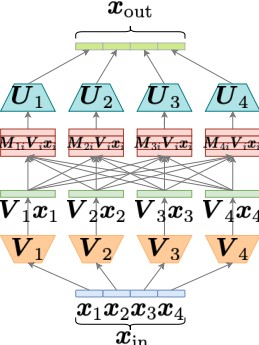

Figure 1: Overview of the NanoMoE Framework, highlighting its key components: input/output partitions, expert matrices $(\boldsymbol{U}_i, \boldsymbol{V}_j)$, and the mixing matrix $(\boldsymbol{M})$.

between the up- and down-projection matrices and reparameterizing these matrices using a sum of Kronecker products. DoRA (yang Liu et al., 2024) reparameterizes the low-rank matrices in LoRA using weight normalization (Salimans & Kingma, 2016). Li et al. (2023) propose approximating a dense weight matrix by the sum of a low-rank matrix and a sparse matrix. Wu et al. (2024) introduce Mixture of LoRA Experts (MoLE), which employs a learnable gating function that utilizes the outputs of multiple LoRAs at each layer to determine composition weights.

## 3  MAIN RESULT

**Low-Rank Factorization Revisited**   To motivate our proposed method, we first revisit the well-established low-rank factorization technique for enhancing parameter efficiency in neural networks. Consider a fully connected layer with weight matrix $\boldsymbol{W} \in \mathbb{R}^{d_2 \times d_1}$, bias vector $\boldsymbol{b} \in \mathbb{R}^{d_2}$, and activation function $\sigma$, where $d_1$ and $d_2$ denote the input and output dimensions, respectively. Let $\boldsymbol{x}_{\text{in}}$ and $\boldsymbol{x}_{\text{out}}$ denote the input and output of this layer. The standard forward pass is given by

$$\boldsymbol{x}_{\text{out}} = \sigma(\boldsymbol{W} \boldsymbol{x}_{\text{in}} + \boldsymbol{b}).$$

The dense weight matrix $\boldsymbol{W}$ contains $d_1 d_2$ parameters.

Low-rank factorization replaces $\boldsymbol{W}$ with the product of two matrices $\boldsymbol{U} \in \mathbb{R}^{d_2 \times r}$ and $\boldsymbol{V} \in \mathbb{R}^{r \times d_1}$, where $r < \min(d_1, d_2)$ is the chosen rank. This yields the modified forward pass:

$$\boldsymbol{x}_{\text{out}} = \sigma(\boldsymbol{U} \boldsymbol{V} \boldsymbol{x}_{\text{in}} + \boldsymbol{b}).$$

This factorization reduces the number of parameters to $(d_1 + d_2)r$, which is significantly less than $d_1 d_2$ (the number of parameters in the dense weight matrix) when the rank $r$ is small enough.

**NanoMoE**   NanoMoE utilizes two matrices $\boldsymbol{U} \in \mathbb{R}^{d_2 \times r}$ and $\boldsymbol{V} \in \mathbb{R}^{r \times d_1}$, each split into $K$ row-wise and column-wise blocks, respectively:

$$\boldsymbol{U} = \begin{pmatrix} \boldsymbol{U}_1^\top & \boldsymbol{U}_2^\top & \cdots & \boldsymbol{U}_K^\top \end{pmatrix}^\top \in \mathbb{R}^{d_2 \times r},$$
$$\boldsymbol{V} = \begin{pmatrix} \boldsymbol{V}_1 & \boldsymbol{V}_2 & \cdots & \boldsymbol{V}_K \end{pmatrix} \in \mathbb{R}^{r \times d_1},$$

where $\boldsymbol{U}_i \in \mathbb{R}^{d_2/K \times r}$ and $\boldsymbol{V}_i \in \mathbb{R}^{r \times d_1/K}$. Similarly, we partition the input vector $\boldsymbol{x}_{\text{in}} \in \mathbb{R}^{d_1}$ and output vector $\boldsymbol{x}_{\text{out}} \in \mathbb{R}^{d_2}$ into $K$ row-wise blocks:

$$\boldsymbol{x}_{\text{in}} = \begin{pmatrix} \boldsymbol{x}_1^\top & \boldsymbol{x}_2^\top & \cdots & \boldsymbol{x}_K^\top \end{pmatrix}^\top,$$
$$\boldsymbol{x}_{\text{out}} = \begin{pmatrix} \boldsymbol{x}_1'^\top & \boldsymbol{x}_2'^\top & \cdots & \boldsymbol{x}_K'^\top \end{pmatrix}^\top,$$

where $\boldsymbol{x}_i \in \mathbb{R}^{d_1/K}$ and $\boldsymbol{x}_i' \in \mathbb{R}^{d_2/K}$.

Each product matrix $\boldsymbol{U}_i \boldsymbol{V}_j \in \mathbb{R}^{d_2/K \times d_1/K}$ acts as an "expert," mapping a block $\boldsymbol{x}_j$ of the input to a corresponding block $\boldsymbol{x}_i'$ of the output. With $K^2$ such experts, we introduce a mixing matrix $\boldsymbol{M}$ to combine their outputs. This matrix is also expressed in block form:

$$\boldsymbol{M} = \begin{pmatrix} \boldsymbol{M}_{11} & \boldsymbol{M}_{12} & \cdots & \boldsymbol{M}_{1K} \\ \boldsymbol{M}_{21} & \boldsymbol{M}_{22} & \cdots & \boldsymbol{M}_{2K} \\ \vdots & \vdots & \ddots & \vdots \\ \boldsymbol{M}_{K1} & \boldsymbol{M}_{K2} & \cdots & \boldsymbol{M}_{KK} \end{pmatrix} \in \mathbb{R}^{Kr \times Kr},$$

where $\boldsymbol{M}_{ij} \in \mathbb{R}^{r \times r}$.

Let $\mathrm{blockdiag}(\boldsymbol{U}_1, \boldsymbol{U}_2, \ldots, \boldsymbol{U}_K)$ denote the block diagonal matrix with $\boldsymbol{U}_i$ on the diagonal. The NanoMoE parameterization is then defined as:

$$\tilde{\boldsymbol{U}} \boldsymbol{M} \tilde{\boldsymbol{V}} \boldsymbol{x}_{\mathrm{in}} = \begin{pmatrix} \sum_{i \in [K]} \boldsymbol{U}_1 \boldsymbol{M}_{1i} \boldsymbol{V}_i \boldsymbol{x}_i \\ \sum_{i \in [K]} \boldsymbol{U}_2 \boldsymbol{M}_{2i} \boldsymbol{V}_i \boldsymbol{x}_i \\ \vdots \\ \sum_{i \in [K]} \boldsymbol{U}_K \boldsymbol{M}_{Ki} \boldsymbol{V}_i \boldsymbol{x}_i \end{pmatrix} \in \mathbb{R}^{d_2}, \tag{1}$$

where

$$\begin{aligned} \tilde{\boldsymbol{U}} &= \mathrm{blockdiag}\left(\boldsymbol{U}_1, \boldsymbol{U}_2, \ldots, \boldsymbol{U}_K\right) \in \mathbb{R}^{d_2 \times Kr}, \\ \tilde{\boldsymbol{V}} &= \mathrm{blockdiag}\left(\boldsymbol{V}_1, \boldsymbol{V}_2, \ldots, \boldsymbol{V}_K\right) \in \mathbb{R}^{Kr \times d_1}. \end{aligned} \tag{2}$$

We illustrate the NanoMoE framework in Fig. 1.

Equation 1 reveals that each block row in the output is a mixture of the outputs of these experts, weighted by the entries of $\boldsymbol{M}$. Specifically, the $i$-th block row is a mixture of the experts $\{\boldsymbol{U}_i \boldsymbol{V}_j \mid j \in [K]\}$.

By inserting $\boldsymbol{M}_{ij}$ between $\boldsymbol{U}_i$ and $\boldsymbol{V}_j$, we enable a more flexible and expressive mixture, enhancing the representation capacity of NanoMoE. While $\boldsymbol{M}$ has shape $Kr \times Kr$, we parameterize it with far fewer parameters to maintain efficiency, as demonstrated in our proposed NanoMoE-I, NanoMoE-II, and NanoMoE-III variants.

- **NanoMoE-I**: Parameterizes $\boldsymbol{M}$ using $K \times K$ parameters (encoded in a matrix $\boldsymbol{A} \in \mathbb{R}^{K \times K}$ with entries $a_{ij}$), where $\boldsymbol{M}_{ij} = a_{ij} \boldsymbol{I}_r$.

- **NanoMoE-II**: Employs $K^2 r$ parameters $\{b_{ijk} \mid i, j \in [K], k \in [r]\}$ to parameterize $\boldsymbol{M}$, with $\boldsymbol{M}_{ij} = \mathrm{diag}(b_{ij})$, where $b_{ij} \triangleq (b_{ij1}, b_{ij2}, \ldots, b_{ijr}) \in \mathbb{R}^r$.

- **NanoMoE-III**: Utilizes $3K^2 r$ parameters $\{c_{ijk} \in \mathbb{R}, \boldsymbol{\alpha}_{ij} \in \mathbb{R}^r, \boldsymbol{\beta}_{ij} \in \mathbb{R}^r \mid i, j \in [K], k \in [r]\}$ to parameterize $\boldsymbol{M}$, with $\boldsymbol{M}_{ij} = \mathrm{diag}(c_{ij}) + \boldsymbol{\alpha}_{ij} \boldsymbol{\beta}_{ij}^\top$.

*Remark* 1. Note that NanoMoE-III generalizes both NanoMoE-II and NanoMoE-I. Specifically, NanoMoE-II can be recovered from NanoMoE-III by setting all $\boldsymbol{\alpha}_{ij}$ and $\boldsymbol{\beta}_{ij}$ to zero. Similarly, NanoMoE-I is a special case of NanoMoE-II where $b_{ijk} = a_{ij}$ for all $i, j \in [K]$ and $k \in [r]$.

Table 1 summarizes the parameter counts for the proposed NanoMoE variants, along with traditional low-rank factorization and fully connected layers. Compared to low-rank factorization, NanoMoE-I, II, and III introduce $K^2$, $K^2 r$, and $3K^2 r$ additional parameters, respectively. In practice, we typically set $K = 2, 4, 8,$ or $16$, which is much smaller than $d_1, d_2,$ and $r$. Therefore, the number of additional parameters is small compared to $(d_1 + d_2)r$, the parameter count for low-rank factorization.

| Parameterization | Number of Parameters |
|---|---|
| Fully Connected | $d_1 d_2$ |
| Low-Rank | $(d_1 + d_2)r$ |
| NanoMoE-I | $(d_1 + d_2)r + K^2$ |
| NanoMoE-II | $(d_1 + d_2)r + K^2 r$ |
| NanoMoE-III | $(d_1 + d_2)r + 3K^2 r$ |

Table 1: The number of parameters of different parameterizations

Theorem 1 below analyzes the expressivity of NanoMoE by examining the space of matrices it can represent. We show that this space is strictly larger than that of low-rank factorization and compute the maximum rank attainable by NanoMoE. Recall the parameter counts for low-rank factorization and NanoMoE summarized in Table 1. For example, compared to low-rank factorization, NanoMoE-I introduces an additional $K^2$ parameters, but achieves a maximum rank $K$ times that of low-rank factorization, as shown in Theorem 1.

**Theorem 1** (Expressivity of NanoMoE, proof in Section 3.1). *Consider the multilinear maps representing the low-rank factorization (LR) and NanoMoE-I parameterizations:*

$$T_{\text{LR}} : \mathbb{R}^{d_2 \times r} \times \mathbb{R}^{r \times d_1} \to \mathbb{R}^{d_2 \times d_1},$$
$$(\boldsymbol{U}, \boldsymbol{V}) \mapsto \boldsymbol{U}\boldsymbol{V},$$
$$T_{\text{NM-I}} : \mathbb{R}^{d_2 \times r} \times \mathbb{R}^{K \times K} \times \mathbb{R}^{r \times d_1} \to \mathbb{R}^{d_2 \times d_1},$$
$$(\boldsymbol{U}, \boldsymbol{A}, \boldsymbol{V}) \mapsto \tilde{\boldsymbol{U}}(\boldsymbol{A} \otimes \boldsymbol{I}_r)\tilde{\boldsymbol{V}},$$

*where $\tilde{\boldsymbol{U}}$ and $\tilde{\boldsymbol{V}}$ are as defined in Equation 2 and $\otimes$ denotes the Kronecker product. Let $\text{im}\,T_{\text{LR}}$ and $\text{im}\,T_{\text{NM-I}}$ denote the images of $T_{\text{LR}}$ and $T_{\text{NM-I}}$, respectively.*

*Then, the following holds:*

*(i) Inclusion: $\text{im}\,T_{\text{LR}} \subseteq \text{im}\,T_{\text{NM-I}}$.*

*(ii) Strict Inclusion: The inclusion is strict, i.e., $\text{im}\,T_{\text{LR}} \subsetneqq \text{im}\,T_{\text{NM-I}}$, if and only if $r < \min\{d_1, d_2\}$ and $K > 1$.*

*(iii) Rank Characterization: In the case of strict inclusion, the maximum ranks attainable by matrices in the two images differ:*

$$\max_{\boldsymbol{W} \in \text{im}\,T_{\text{LR}}} \text{rank}(\boldsymbol{W}) = r,$$
$$\max_{\boldsymbol{W} \in \text{im}\,T_{\text{NM-I}}} \text{rank}(\boldsymbol{W}) = \min\{d_1, d_2, Kr\} > r.$$

*Remark* 2. Theorem 1 (specifically, Item iii) establishes a clear separation between the maximum rank attainable by low-rank factorization (which is $r$) and that attainable by NanoMoE-I (which is $\min\{d_1, d_2, Kr\}$). When $r$ is small enough to ensure $Kr < \min\{d_1, d_2\}$, this signifies a potential $K$-fold increase in the maximum attainable rank due to the NanoMoE-I parameterization.

*Remark* 3. Since NanoMoE-I is a special case of NanoMoE-II and NanoMoE-III (Remark 1), denoting the images of the NanoMoE-II and NanoMoE-III parameterizations by $\text{im}\,T_{\text{NM-II}}$ and $\text{im}\,T_{\text{NM-III}}$ respectively, we have the following chain of inclusions:

$$\text{im}\,T_{\text{LR}} \subseteq \text{im}\,T_{\text{NM-I}} \subseteq \text{im}\,T_{\text{NM-II}} \subseteq \text{im}\,T_{\text{NM-III}}.$$

Furthermore, if $r < \min\{d_1, d_2\}$ and $K > 1$, the inclusions $\text{im}\,T_{\text{LR}} \subseteq \text{im}\,T_{\text{NM-II}}$ and $\text{im}\,T_{\text{LR}} \subseteq \text{im}\,T_{\text{NM-III}}$ are strict. Moreover, the maximum rank attainable by matrices in $\text{im}\,T_{\text{NM-II}}$ and $\text{im}\,T_{\text{NM-III}}$ is also $\min\{d_1, d_2, Kr\}$.

### 3.1 PROOF OF THEOREM 1

*Proof of Theorem 1.* **Proof of Item i.** The inclusion $\text{im}\,T_{\text{LR}} \subseteq \text{im}\,T_{\text{NM-I}}$ is straightforward. Setting $\boldsymbol{A} = \boldsymbol{1}_{K \times K}$ (the all-ones matrix), we have

$$T_{\text{NM-I}}(\boldsymbol{U}, \boldsymbol{A}, \boldsymbol{V}) = \boldsymbol{U}\boldsymbol{V} = T_{\text{LR}}(\boldsymbol{U}, \boldsymbol{V}).$$

Hence, $\text{im}\,T_{\text{LR}} \subseteq \text{im}\,T_{\text{NM-I}}$.

**Proof of Item ii ("only if" part).** Next, we establish that if $r \geq \min\{d_1, d_2\}$ or $K = 1$, then $\text{im}\,T_{\text{LR}} = \text{im}\,T_{\text{NM-I}}$.

Case 1: $K = 1$. In this case, $\boldsymbol{A}$ reduces to a scalar $a$, and $\tilde{\boldsymbol{U}} = \boldsymbol{U}$, $\tilde{\boldsymbol{V}} = \boldsymbol{V}$. Consequently, $\tilde{\boldsymbol{U}}(\boldsymbol{A} \otimes \boldsymbol{I}_r)\tilde{\boldsymbol{V}} = a\boldsymbol{U}\boldsymbol{V}$, implying $\text{im}\,T_{\text{LR}} = \text{im}\,T_{\text{NM-I}}$.

Case 2: $r \geq \min\{d_1, d_2\}$. The rank of any matrix in $\text{im}\,T_{\text{NM-I}}$ is bounded above by $\min\{d_1, d_2\}$. Since $r \geq \min\{d_1, d_2\}$, for any $\boldsymbol{W} \in \text{im}\,T_{\text{NM-I}}$ there exist matrices $\boldsymbol{U} \in \mathbb{R}^{d_2 \times r}$ and $\boldsymbol{V} \in \mathbb{R}^{r \times d_1}$ such that $\boldsymbol{W} = \boldsymbol{U}\boldsymbol{V}$. This implies $\boldsymbol{W} \in \text{im}\,T_{\text{LR}}$, and hence $\text{im}\,T_{\text{NM-I}} \subseteq \text{im}\,T_{\text{LR}}$. The reverse inclusion $\text{im}\,T_{\text{LR}} \subseteq \text{im}\,T_{\text{NM-I}}$ has already been established, so we conclude $\text{im}\,T_{\text{LR}} = \text{im}\,T_{\text{NM-I}}$.

**Proof of Item ii ("if" part) and Item iii.** To show that this inclusion is strict under the assumptions $r < \min\{d_1, d_2\}$ and $K > 1$, we will prove $\operatorname{im} T_{\text{LR}} \neq \operatorname{im} T_{\text{NM-I}}$. Choose full-rank matrices $\boldsymbol{U}_1, \ldots, \boldsymbol{U}_K \in \mathbb{R}^{d_2/K \times r}$ and $\boldsymbol{V}_1, \ldots, \boldsymbol{V}_K \in \mathbb{R}^{r \times d_1/K}$, and a full-rank matrix $\boldsymbol{A} \in \mathbb{R}^{K \times K}$. We then have:

$$\operatorname{rank}(\tilde{\boldsymbol{U}}) = \sum_{i \in [K]} \operatorname{rank}(\boldsymbol{U}_i) = \min\{d_2, Kr\},$$

$$\operatorname{rank}(\tilde{\boldsymbol{V}}) = \sum_{i \in [K]} \operatorname{rank}(\boldsymbol{V}_i) = \min\{d_1, Kr\},$$

$$\operatorname{rank}(\boldsymbol{A} \otimes \boldsymbol{I}_r) = \operatorname{rank}(\boldsymbol{A}) \operatorname{rank}(\boldsymbol{I}_r) = Kr.$$

By Sylvester's rank inequality:

$$\operatorname{rank}(\tilde{\boldsymbol{U}}(\boldsymbol{A} \otimes \boldsymbol{I}_r))$$
$$\geq \operatorname{rank}(\tilde{\boldsymbol{U}}) + \operatorname{rank}(\boldsymbol{A} \otimes \boldsymbol{I}_r) - Kr$$
$$= \min\{d_2, Kr\}.$$

We now demonstrate that

$$\operatorname{rank}(\tilde{\boldsymbol{U}}(\boldsymbol{A} \otimes \boldsymbol{I}_r)\tilde{\boldsymbol{V}}) \geq \min\{d_1, d_2, Kr\}.$$

Case 1: $Kr \geq \max\{d_1, d_2\}$. In this case, $\operatorname{rank}(\tilde{\boldsymbol{U}}(\boldsymbol{A} \otimes \boldsymbol{I}_r)) = \min\{d_2, Kr\} = Kr$. Since $\tilde{\boldsymbol{V}}$ is full-rank with rank $\min\{d_1, Kr\} = Kr$, the product $\tilde{\boldsymbol{U}}(\boldsymbol{A} \otimes \boldsymbol{I}_r)\tilde{\boldsymbol{V}}$ is also full-rank and has rank $\min\{d_1, d_2\} = \min\{d_1, d_2, Kr\}$.

Case 2: $Kr < \max\{d_1, d_2\}$. By Sylvester's rank inequality:

$$\operatorname{rank}(\tilde{\boldsymbol{U}}(\boldsymbol{A} \otimes \boldsymbol{I}_r)\tilde{\boldsymbol{V}})$$
$$\geq \operatorname{rank}(\tilde{\boldsymbol{U}}(\boldsymbol{A} \otimes \boldsymbol{I}_r)) + \operatorname{rank}(\tilde{\boldsymbol{V}}) - Kr$$
$$= \min\{d_2, Kr\} + \min\{d_1, Kr\} - Kr$$
$$= \min\{\max\{d_1, d_2\}, Kr\}$$
$$\quad + \min\{\min\{d_1, d_2\}, Kr\} - Kr$$
$$= Kr + \min\{d_1, d_2, Kr\} - Kr$$
$$= \min\{d_1, d_2, Kr\}.$$

Since $r < \min\{d_1, d_2\}$ and $K > 1$, it follows that

$$\operatorname{rank}(\tilde{\boldsymbol{U}}(\boldsymbol{A} \otimes \boldsymbol{I}_r)\tilde{\boldsymbol{V}}) \geq \min\{d_1, d_2, Kr\} > r.$$

Since $\tilde{\boldsymbol{U}} \in \mathbb{R}^{d_2 \times Kr}$ and $\tilde{\boldsymbol{V}} \in \mathbb{R}^{Kr \times d_1}$, we have $\operatorname{rank}(\tilde{\boldsymbol{U}}(\boldsymbol{A} \otimes \boldsymbol{I}_r)\tilde{\boldsymbol{V}}) \leq \min\{d_1, d_2, Kr\}$. Combining this with the previously established lower bound, we conclude that

$$\operatorname{rank}(\tilde{\boldsymbol{U}}(\boldsymbol{A} \otimes \boldsymbol{I}_r)\tilde{\boldsymbol{V}}) = \min\{d_1, d_2, Kr\}.$$

As all matrices in $\operatorname{im} T_{\text{LR}}$ have rank at most $r$, we conclude that $\operatorname{im} T_{\text{LR}} \neq \operatorname{im} T_{\text{NM-I}}$.

$\square$

## 4 EXPERIMENT

Our initial experiments focus on the first fully connected layer of the OPT-13B model (Zhang et al., 2022), whose weight matrix has shape $(20480, 5120)$, corresponding to $d_1 = 20480$ and $d_2 = 5120$ (due to PyTorch's convention of left-multiplying the input by the weight matrix). To simulate training, we generate 100,000 samples of dimension 20480, each entry drawn from a normal distribution with standard deviation 5. The dataset is split into 75% for training and 25% for testing.

We fit this dataset using low-rank factorization and the three NanoMoE variants, varying $K \in \{2, 4, 8, 16, 32, 64, 128\}$ and $r \in \{2560, 1280, 640, 320, 160, 80, 40\}$. We record training loss, test

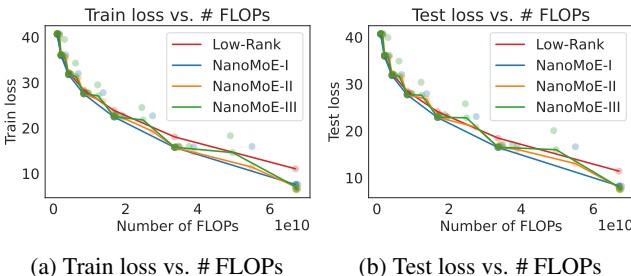

(a) Train loss vs. # FLOPs    (b) Test loss vs. # FLOPs

Figure 2: Comparison of training and test loss vs. FLOPs for Low-Rank Factorization and NanoMoE Variants on the first fully connected layer of OPT-13B. Lower envelope lines represent the optimal parameter choices for each model.

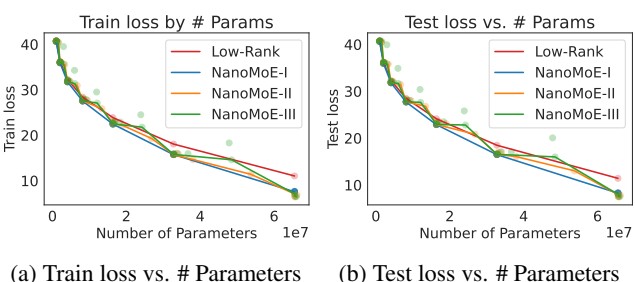

(a) Train loss vs. # Parameters    (b) Test loss vs. # Parameters

Figure 3: Comparison of training and test loss vs. the number of parameters for Low-Rank Factorization and NanoMoE Variants on the first fully connected layer of OPT-13B. Lower envelope lines represent the optimal parameter choices for each model.

loss, floating point operations (FLOPs) (computed via `numpy.einsum_path`), and parameter counts. Figures 2 and 3 plot the results for all $(K, r)$ combinations, with lines connecting data points on the lower envelope of each model's performance.

The data points above these lines reflect suboptimal choices of $K$ and $r$. For example, some combinations use an unnecessarily large $r$ to achieve a given train/test loss, while a smaller $r$ would suffice. The lower envelope lines thus represent optimal $(K, r)$ pairings for each model, enabling a fair comparison. Notably, Figures 2 and 3 reveal that for a fixed FLOP budget or parameter budget, the NanoMoE variants consistently outperform low-rank factorization in terms of both training and test loss.

We conduct a second set of experiments on the AG News classification dataset (Zhang et al., 2015). This dataset comprises 120,000 training examples and 7,600 test examples, and we utilize the original train/test split provided. The neural network architecture of the experiments on the AG News classification dataset consists of the following layers:

- Text vectorization layer with output sequence length of 250.
- Embedding layer with embedding dimension of 300.
- 1D global average pooling layer.
- Low-rank factorization layer or NanoMoE layer (depending on the experiment).
- Final fully-connected layer that outputs a 4-dimensional vector for classification.

We evaluate different hyperparameter configurations for both NanoMoE and low-rank factorization. We sweep over $K$ in the range $[2, 150]$ and $r$ in the range $[2, 300]$. Figures 4 and 5 plot the results for all $(K, r)$ combinations, with lines connecting data points on the lower envelope of each model's performance. Consistent with the observations from the first experiment set (refer to Figures 2 and 3), the second set of experiments on the AG News dataset reveals an even wider gap between the training/test loss curves of low-rank factorization and those of the NanoMoE variants. Among the NanoMoE variants, NanoMoE-I achieves the best overall performance in terms of loss.

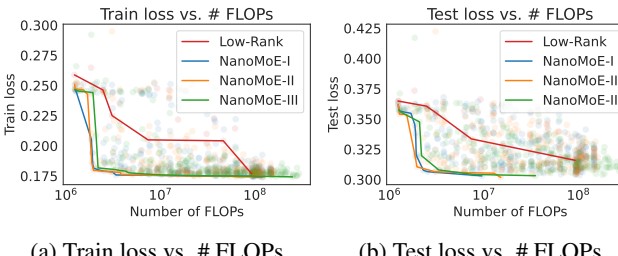

(a) Train loss vs. # FLOPs     (b) Test loss vs. # FLOPs

Figure 4: Comparison of training and test loss vs. FLOPs for Low-Rank Factorization and NanoMoE Variants on the AG News classification dataset. Lower envelope lines represent the optimal parameter choices for each model.

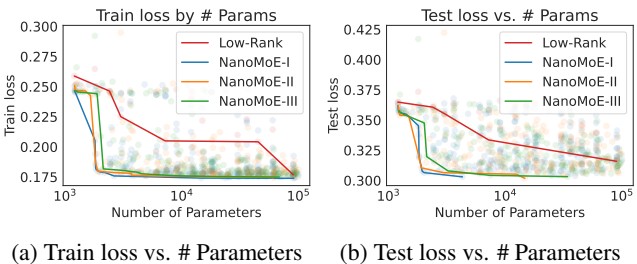

(a) Train loss vs. # Parameters     (b) Test loss vs. # Parameters

Figure 5: Comparison of training and test loss vs. the number of parameters for Low-Rank Factorization and NanoMoE Variants on the AG News classification dataset. Lower envelope lines represent the optimal parameter choices for each model.

## 5 CONCLUSION

This work introduces NanoMoE, a novel parameter-efficient building block designed to replace fully-connected layers and low-rank factorization layers in neural networks. We theoretically demonstrate that NanoMoE offers strictly greater expressivity compared to low-rank factorization, while requiring only a minimal increase in parameters. Furthermore, our empirical results consistently validate that NanoMoE achieves superior performance in terms of both training and test loss across various FLOPs budgets and parameter constraints. These findings suggest that NanoMoE presents a promising avenue for developing more efficient and effective neural network architectures.

## 6 FUTURE WORK

Our study presents several opportunities for future work. First, while our experiments showcase the parameter efficiency of NanoMoE, exploring principled methods for selecting the optimal hyperparameters $K$ (number of experts) and $r$ is crucial to maximize this efficiency. Second, we haven't investigated the performance of NanoMoE within the context of LoRA-type fine-tuning (Hu et al., 2022). Additionally, exploring NanoMoE's potential in pre-training large language models and employing stacked NanoMoE architectures (e.g., replacing all fully-connected layers with NanoMoE layers) are promising avenues for future research.

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
