# OpenReview forum: "NanoMoE: Scaling Mixture of Experts to Individual Layers for Parameter-Efficient Deep Learning"
_ICLR.cc/2025/Conference — Submitted to ICLR 2025_

### Official Review · Reviewer_cSpB · 2024-10-27

**Soundness:** 2
**Presentation:** 3
**Contribution:** 2
**Rating:** 3
**Confidence:** 4

**Summary:**

The authors propose to extend low-rank approximation of standard neural network weight matrices of the form W=UV into W = blockdiag(U) M blockdiag(V) where blockdiag(U) is a block diagonal reshaping of the original matrix U. M is a block matrix interpreted as expert weights for each possible combinations of subblocks from blockdiag(U) and blockdiag(V). The M matrix is parametrized in three different ways (scalar times identity, diagonal, diagonal plus outer product) with increasing expressivity proved theoretically but also increasing computational cost. The authors empirically validate that the proposed approach is better than low-rank in terms of train/test loss for 1) a synthetic data setting when controlling parameters and FLOPs, 2) AG news classification when controlling for parameters and FLOPs.

**Strengths:**

- The idea is quite relevant to lots of on-going works that replace dense matrices with different structure matrices for improved performance. I think it’s a nice addition to the community.
- The paper is well-written and easy to follow

**Weaknesses:**

The primary problem with all of the empirical evaluations in this paper is that they are not informative about whether the proposed approach is actually a good replacement for standard MoE layers or not. The baseline is just low-rank, which is shown to be less expressive already compared to the proposed nanoMoE. It’s essential to compare to a standard MoE where you’re not using any low-rank structure but with just dense matrices.

It’s also not surprising that when controlling for parameters, nanoMoE is performing better than low-rank since the parameter overhead introduced by K and r are relatively small (the authors sweeped over small values of K).

I’m willing to change my scores if the authors add the dense matrix $W\in\mathbb{R}^{d_2\times d_1}$ baseline and the standard MoE with dense matrices baseline, at least in a limited setting if the compute budget is a problem during rebuttal.

**Questions:**

What’s the loss function in the synthetic dataset experiments where you are sampling i.i.d gaussian random vectors of dimension 20480? The text mentions that it’s testing the FC layer from OPT-13B, which is a language model. It’s not clear to me what’s the training objective function here.

---

### Official Review · Reviewer_tHzT · 2024-10-27

**Soundness:** 3
**Presentation:** 3
**Contribution:** 2
**Rating:** 3
**Confidence:** 4

**Summary:**

The paper introduces a novel variant of the mixture of experts model aimed at reducing the number of parameters and floating-point operations (FLOPs) in neural networks. This is achieved by factorizing individual feedforward layers and their corresponding input feature maps into smaller chunks, and then aggregating their outputs. By applying this operation to dense layers, the method significantly reduces parameter count and memory requirements while maintaining competitive performance levels

The paper’s main contributions include: introducing NanoMoE, a parameter-efficient block family with three complexity levels (NanoMoE-I, II, and III); proving NanoMoE’s higher expressivity over low-rank factorization with minimal parameter increase; and validating through experiments that NanoMoE achieves better model quality than low-rank factorization at similar resource budgets.

**Strengths:**

The strengths of this paper can be summarized as follows:

- Efficiency in Resource Usage: The proposed method effectively reduces the number of parameters and computational demands,  making it suitable for deployment in resource-constrained environments.
- Maintained Performance: Despite the reduction in computational resources, the model achieves results that are competitive with more resource-intensive approaches.
- Innovative Approach: The factorization and aggregation technique offers a fresh perspective on optimizing neural network architectures.

**Weaknesses:**

Although the idea is interesting, the proposed method has several major weaknesses:

- **Lack of Inference Speed Evaluation**: While the main objective is to reduce computational cost and memory footprint, the experiments focus primarily on parameter reduction. There is no discussion of whether the method improves inference speed, which is critical for assessing practical efficiency gains.

- **Limited Experimental Scope**: The authors conduct only two experiments on a single dense layer or a simple model, making it difficult to assess the method’s feasibility for real-world deployment and its performance in more complex scenarios.

- **Narrow Evaluation Metrics**: The evaluation is limited to loss reduction without considering classification accuracy, which would be valuable for classification tasks. Including transfer learning experiments would further help to gauge the method’s effectiveness across tasks.

- **Absence of Baseline Comparison**: The approach of weight partitioning and non-gated mixtures of experts is not new[1].  Comparisons with existing methods that use similar techniques, such as [1] focusing on parameter reduction, would provide clearer insights into the proposed method’s relative performance and innovation.

[1] Scaling Laws for Fine-Grained Mixture of Experts

**Questions:**

**Questions:**
- How were the hyperparameters chosen? Was any analysis conducted to determine optimal values, especially for selecting the dense layer in synthetic data experiments?
- What prevented the focus from extending to multiple FFN layers? Was this due to increased complexity, as each dense layer would require a similar setup?
- Why has NanoMoE not been tested on more complex architectures beyond single dense layers? How do the authors envision scaling it for larger models?
- Is there a reason NanoMoE did not incorporate sparse gating mechanisms, as seen in other MoE frameworks?
- How does NanoMoE compare with other parameter-efficient MoE-based or low-rank models in terms of accuracy and parameter reduction? Were any qualitative comparisons made?
- Has NanoMoE been tested in transfer learning contexts? Would it retain its efficiency and performance when adapted to new tasks?


**Suggestions:**
The theoretical foundation is strong, but more experiments are needed to assess NanoMoE's performance and complexity compared to other MoE and existing approaches. I suggest:
- Adding performance comparisons with some existing baselines.
- Extending the experiments to more layers beyond the embedding layer, ideally including FFN and attention layers for a thorough evaluation.
- Compared with other MoE frameworks, NanoMoE’s structure is similar and would benefit from these benchmarks.

---

### Official Review · Reviewer_53vK · 2024-10-29

**Soundness:** 2
**Presentation:** 1
**Contribution:** 2
**Rating:** 3
**Confidence:** 4

**Summary:**

The paper introduces NanoMoE, a building block which adds an additional mixing layer to low-rank factorization layers for linear projections. The paper draws connections to the mixing matrix from the mixture of expert literature, and characterises the space of matrices it can represent. Finally the authors test the proposed method on a synthetic task on various FLOPs budgets and on the AG news classification task.

**Strengths:**

- The method to mix low-rank factorization intermediate outputs is interesting.
- The paper emphasizes how performance improves as a function of FLOPs, which is important to develop scalable methods, especially for pre-training.

**Weaknesses:**

**Conceptual Framing**

- The connection to the sparsely-gated mixture of experts literature is very weak. As the method is described in section 3, the matrix M performs mixing over the partitioned input x_in, this is very different from [1] which is referenced in section 1, where specific components of the network learn to route inputs to “experts”. Nanomoe rather does a sort of sparse mixing over the embedding dimension, where no sparse routing or “expertise” is learned.
- The claims in the paper are too overreaching.
    - Mixture of expert layers and sparse layers have already been applied to individual layers in prior work, see [1] and [2] [3] for preliminary references, a more thorough literature review should be in the paper. This work does not scale more than previous work in terms of applying them to whole components of the network, or in the experimental setting size (which is very modest in NanoMoE).
    - Section 1 says “We formally define this problem as parameter-efficient deep learning”. There is not a formal definition of this in the paper, just a formal definition of the proposed method. Also, more related to the proposed approach, is sparse deep learning, which has a rich body of literature. The paper hardly proposes a new problem that is not already known or tackled in the deep learning literature.
    - There is no discussion about hardware efficiency other than a very loose definition of FLOPs in numpy. Real-world hardware efficiency is necessary to scale up methods as implied in the title.
- The Monarch matrices line of work [5] [6] [7] seems very relevant to this work (it is not cited), as it deals with a more efficient building blocks with block-diagonal matrices, with detailed discussion on expressiveness, efficiency, experimental work and covering a super-set of the scope of this paper (both pre training with mixed sparse+dense, and finetuning of dense models). I highly recommend the authors to review [5] as a blueprint for this work. It’s worth a discussion of the differences between both methods, both in terms of modelling and hardware efficiency for pretraining; at the very least, this seems like an important baseline to have in the experimental section.

**Experiments**

- I found the first experiment from the OPT-13b layer very confusing. There is no description about what loss is being optimised, which makes it very difficult to interpret the results — the losses in Figure 2) and b) seem high but without any description it is not possible to know if any of the models is learning anything useful at all. Moreover, the input is random gaussians with a rather high standard deviation, again without any explanation, this task does not seem to be representative of a real training task at all.
- The experiments compare to Low-Rank training as a baseline. However, a more important comparison to do is with a fully dense layer, which is what actually is commonly used in pre-training (which the paper advocates for in Section 1). Also, the related work section describes a number of models that would be important baselines to compare to, low-rank is arguably a simple baseline and not SOTA.
- For the AG News classification dataset, there’s several important experimental details missing, which are crucial to understand the empirical merit of NanoMoE:
    - What is the loss being optimised?
    - What are the details of the vectorization layer? What’s the vocabulary size? How are words out of vocabulary handled?
    - How many epochs/steps occur during training?
    - What is the optimizer and what hyper-parameters are used? (batch size, learning rate, regularisation, etc)
    - How are the weights initialised in the NanoMoE layers? More generally speaking, which hyper-parameters are different in NanoMoE vs the low-rank baseline?
    - What is the granularity of the K and r ranges?
    - What is the activation function used in the experiments?
- [7] shows that a careful parametrization is needed for structured matrices. This is a missing detail on the hyper-parameters, but also a missing discussion for NanoMoE too.
- Figures 4) and 5) are hard to visualise with all the data points being very transparent. There is a lot of variance per Flop Budget, which probably is due to interactions of K and r. It is important to disentangle these effects as well.
- Plotting the low envelope seems to ignore the fact that NanoMoE is overfitting at higher FLOP counts on figure 5b (if that’s not the case, the colours are making this difficult to interpret). Is NanoMoE more prone to overfitting at higher FLOP budgets? If it is, then the method is not very promising, it could also be a lack of proper regularisation, but this is not clear given the lack of experimental details.
- Modern NLP solves classification problems such as AG News with unsupervised pre-training + transfer-learning (BERT-style models) or few-shot learning (GPT-style models). While large-scale pre-training is very expensive, there is work to pretrain BERT-style models in as little as 1 GPU day [4] which is more suitable to academic budgets. A *single* and small-scale experiment on this unsupervised learning setup, would be more apt to compare to modern methods in NLP (this can very well be the single best combination from the AG news experiment).
- The definition of FLOPs seem to focus on inference considerations, as I think it computes the output of numpy.einsum_path over a single einsum operation (is not clear what operations are included in the call to enisum_path, a spelled out code snippet would be useful). However, this paper focuses as per section 1 on efficient pre-training. This calls for a definition of FLOPs per training step, which includes: forward and backward FLOPs, runtime bounds such as given in [5], and practical step time on modern accelerators. A number of these can be future work, but it needs to be disclosed explicitly in order to consider the merits of the paper.
- All in all, I consider the experimental section to be too weak to claim this in the conclusion: “our empirical results consistently validate that NanoMoE achieves superior performance”. More thorough experiments need to be done before claiming this.

[1] https://openreview.net/forum?id=B1ckMDqlg

[2] https://proceedings.mlr.press/v162/dao22a/dao22a.pdf

[3] https://openreview.net/forum?id=-b5OSCydOMe

[4] https://proceedings.mlr.press/v202/geiping23a/geiping23a.pdf

[5] https://proceedings.mlr.press/v162/dao22a/dao22a.pdf

[6] https://openreview.net/forum?id=cB0BImqSS9&noteId=98BZANkxc8

[7] https://proceedings.mlr.press/v235/qiu24f.html

**Questions:**

- What are the hyper-parameters used to conduct the experiments? See weaknesses section for what it’s relevant to discuss. What’s the sensitivity of NanoMoE to hyper-parameters?
- What is the loss function used to optimise the first experiment of section 4? What is this experiment trying to show (irrespective of matching conclusions with the AG news experiment)?
- What is the runtime of NanoMoE compared to dense matmuls either with low-rank or not? How complicated is it to run this efficiently in modern accelerators? Is this future work?
- Is NanoMoE more prone to overfitting in the experiments?

---

### Official Review · Reviewer_P46a · 2024-11-04

**Soundness:** 2
**Presentation:** 3
**Contribution:** 2
**Rating:** 3
**Confidence:** 4

**Summary:**

This paper introduces NanoMoE, a novel family of structured matrices that achieves superior flexibility compared to low-rank matrices with minimal increase in parameters or FLOPs. The paper theoretically proves that NanoMoE can have a significantly higher rank and is strictly more flexible than low-rank matrices for similar parameter counts. Some experiments confirm the improved performance of NanoMoE layers relative to low-rank layers.

**Strengths:**

- NanoMoE is a novel family of structured matrices with clear theoretical advantages over low-rank matrices in terms of expressiveness, especially in achieving higher ranks for the same parameter count.
- The paper rigorously proves the said advantages of NanoMoE
- Experiments support the theory.

**Weaknesses:**

- Experiments are only done on toy problems such as dense matrix approximation and a small text classification dataset. Can the authors present experiments on tasks such as image classification on CIFAR-10 / ImageNet and language modeling (e.g., using the nanoGPT codebase)? Results on these benchmarks have been used in evaluating new structured matrices in recent works [1, 2].
- There are already many equally parameter-efficient structured matrices that have the advantage of being full-rank, such as the Kronecker product, Tensor-Train decomposition, and Monarch matrices [1]. There is no comparison with these alternatives.
- While more expressive than the usual low-rank matrix, I believe NanoMoE will require more memory to store the activations (intermediate tensors have size $K r$ rather than just $r$). Moreover, I suspect the tensor core utilization will be lower because the block diagonal matrices involve contraction with smaller ranges, resulting in worse wall clock times despite having a minimal increase in FLOPs. The authors did not discuss these potential limitations.
- The experiment section does not provide details about how the models were trained. For example, are the learning rates well-tuned? Prior work [2, 3] has shown that structured matrices require very different learning rates than those commonly used for dense layers, making a well-tuned learning rate important for a fair comparison.
- The paper presents the connection to MoE as a strength since it has been shown to be more compute-efficient for pre-training LLMs. But only sparse MoE models have demonstrated improved training efficiency and is what was used in referenced models such as Mixtral and Switch Transformer. The proposed NanoMoE, however, is not a sparse MoE model and is, therefore, unlikely to lead to similar benefits. The authors carefully discuss this distinction.
- Recent works have used structured matrices to build MoE in each linear layer, similar to what is proposed in this work. I suggest the authors to discuss these highly related works. [3, 4]

[1] Dao et al. 2022. Monarch: Expressive Structured Matrices for Efficient and Accurate Training

[2] Qiu et al. 2024. Compute Better Spent: Replacing Dense Layers with Structured Matrices

[3] Potapczynski et al. 2024. Searching for Efficient Linear Layers over a Continuous Space of Structured Matrices

[4] Oldfield et al. 2024. Multilinear Mixture of Experts: Scalable Expert Specialization through Factorization

**Questions:**

- Could you elaborate on the training details, such as the learning rate and optimizer? Did you properly tune them and were the results sensitive to these choices?
- Does NanoMoE lead to higher activation memory due to larger intermediate tensors?
- How does NanoMoE compare with low-rank in terms of performance vs wall clock time (rather than parameter count or FLOPs) ?

---

### Meta-Review · Area_Chair_3Jcn · 2024-12-20

**Metareview:**

**Summary**

This paper presents NanoMoE, a new class of structured matrices that offers enhanced flexibility over low-rank matrices while minimally increasing the number of parameters or FLOPs. Theoretical evidence demonstrates that NanoMoE can achieve a significantly higher rank and provides greater flexibility than low-rank matrices for comparable parameter counts. Experimental results on small scale problems further confirm that NanoMoE layers outperform low-rank layers in terms of performance.

**Strengths**

The reviewers unanimously highlighted several strengths of the proposed framework:
* NanoMoE is a novel family of structured matrices with clear theoretical advantages over low-rank matrices in terms of expressiveness, especially in achieving higher ranks for the same parameter count.  This factorization and aggregation technique offers a fresh perspective on optimizing neural network architectures.
* The paper rigorously proves the said advantages of NanoMoE
* Experiments support the theory and show performance improves as a function of FLOPs, which is important to develop scalable methods, especially for pre-training.

**Weaknesses**

Several core weaknesses was brought up by the reviewers. These include:
* Although the primary goal is to reduce computational cost and memory usage, the experiments focus mainly on parameter reduction. The lack of discussion on whether the method improves inference speed limits the assessment of its practical efficiency gains.
* The authors conduct only two experiments on a single dense layer or a simple model, which makes it challenging to evaluate the method’s applicability to real-world deployment and its performance in more complex scenarios.

**Conclusion**

The majority of reviewers acknowledge the merits of the paper but criticize the experimental setup as rudimentary and inconclusive. Unfortunately, the authors did not submit a rebuttal. Given the unanimous consensus among the reviewers favoring rejection, I also vote to reject this paper.

**Additional Comments On Reviewer Discussion:**

Given the unanimous vote of the reviewers and the lack of responses from the authors, we did not find it necessary to further discuss this paper.

---

### Decision · Program_Chairs · 2025-01-22

Reject